# Midline Meningiomas of the Anterior Skull Base: Surgical Outcomes and a Decision-Making Algorithm for Classic Skull Base Approaches

**DOI:** 10.3390/cancers12113243

**Published:** 2020-11-03

**Authors:** Amir Kaywan Aftahy, Melanie Barz, Philipp Krauss, Arthur Wagner, Nicole Lange, Alaa Hijazi, Benedikt Wiestler, Bernhard Meyer, Chiara Negwer, Jens Gempt

**Affiliations:** 1Department of Neurosurgery, Klinikum Rechts der Isar, School of Medicine, Technical University Munich, 81675 Munich, Germany; Melanie.barz@tum.de (M.B.); philipp.krauss@tum.de (P.K.); arthur.wagner@tum.de (A.W.); Nicole.Lange@tum.de (N.L.); alaa.h@hotmail.de (A.H.); Bernhard.Meyer@tum.de (B.M.); Chiara.Negwer@tum.de (C.N.); Jens.Gempt@tum.de (J.G.); 2Department of Neuroradiology, Klinikum Rechts der Isar, School of Medicine, Technical University Munich, 81675 Munich, Germany; B.Wiestler@tum.de

**Keywords:** olfactory groove meningioma, planum sphenoidale meningioma, tuberculum sellae meningioma, anterior skull base, operative technique, neurosurgical oncology, transcranial approaches

## Abstract

**Simple Summary:**

Resectioning midline meningiomas of the anterior skull base such as olfactory groove, planum sphenoidale, or tuberculum sellae is challenging, and determining the appropriate approach is important. Based on our experience with midline meningiomas, we propose a decision algorithm for choosing suitable transcranial approaches. With dichotomizing classic skull bases approaches into median and lateral ones, we display that median approaches provide satisfactory results for olfactory groove meningiomas, whereas lateral approaches enable sufficient exposure of the visual apparatus for planum sphenoidale meningiomas or tuberculum sellae meningiomas. This manuscript aims to point out the sufficiency and feasibility of classic transcranial techniques.

**Abstract:**

(1) Background: Midline meningiomas such as olfactory groove (OGMs), planum sphenoidale (PSMs), or tuberculum sellae meningiomas (TSMs) are challenging, and determining the appropriate approach is important. We propose a decision algorithm for choosing suitable transcranial approaches. (2) Methods: A retrospective chart review between 06/2007 and 01/2020. Clinical outcomes, radiographic findings, and postoperative complication rates were analyzed with respect to operative approaches. (3) Results: We included 88 patients in the analysis. Of these, 18.2% (16/88) underwent an interhemispheric approach, 72.7% (64/88) underwent a pterional/frontolateral/supraorbital approach, 2.3% (2/88) underwent a unilateral subfrontal approach, and 6.8% (6/88) underwent a bifrontal approach. All OGMs underwent median approaches, along with one PSM. All of the other PSMs and TSMs were resected via lateral approaches. The preoperative tumor volume was ∅20.2 ± 27.1 cm^3^. Median approaches had significantly higher tumor volume but also higher rates of Simpson I resection (75.0% vs. 34.4%). An improvement of visual deficits was observed in 34.1% (30/88). The adverse event rate was 17.0%. Median follow-up was 15.5 months (range 0–112 months). (4) Conclusions: Median approaches provides satisfying results for OGMs, lateral approaches enable sufficient exposure of the visual apparatus for PSMs and TSMs. In proposing a simple decision-making algorithm, the authors found that satisfactory outcomes can be achieved for midline meningiomas.

## 1. Introduction

Meningiomas of the anterior skull base are complex lesions due to local bone invasion and the invasion of adjacent neural and vascular structures. Skull base meningiomas represent 25% of all meningiomas [1]. Meningiomas of the midline anterior skull base include tuberculum sellae meningiomas (TSMs) and planum sphenoidale meningiomas (PSMs), representing 5–10% of all intracranial meningiomas, and olfactory groove meningiomas (OGMs), representing 8–13% (3–7). The treatment of choice is maximal safe resection to achieve a low Simpson grading, which is still the main predictor for recurrence-free survival in contemporary studies [2,3]. Other treatment modalities include radiation (including proton beam therapy) and devascularization via catheter embolization. Various classic approaches such as the pterional, frontolateral, supraorbital, subfrontal, bifrontal, and interhemispheric approaches have been described as achieving an optimal visualization of the situs [1,3,4,5,6,7,8,9]. In addition to these classical approaches, more complex skull-base approaches have been popularized in recent years, including the use of neuroendoscopy [3,9,10].

In this manuscript, we investigate the extent of resection along with the clinical outcomes and adverse events in a large cohort of patients with midline meningiomas of the anterior skull base at a single tertiary center who underwent classic transcranial approaches.

## 2. Results

### 2.1. Clinical Baseline Characteristics

In total, 88 patients (62 (70.4%) female, 26 (29.5%) male) fulfilled the inclusion criteria and were analyzed. Olfactory dysfunction and psychomotor decline before surgery were present significantly often in patients with OGMs, as compared to patients with other midline meningiomas of the anterior skull base. Patients with TSM showed significantly better preoperative Karnofsky Performance Status Scale (KPSS) than patients with the other entities did, with overall high levels of KPSS but significantly higher rates of visual affection (Table 1).

### 2.2. Tumor Characteristics and Approach-Related Findings

Histopathological analysis showed a WHO I meningioma in 81 cases (92%), a WHO II meningioma in six (7%) cases, and a WHO III meningioma in one case (1%). The preoperative tumor volume was ∅20.2 ± 27.1 cm^3^ (OGM ∅26.6 ± 26.2 cm^3^; PSM ∅17.6 cm^3^ ± 32.8 cm^3^; TSM ∅2.6 cm^3^ ± 1.9 cm^3^). Patients with OGMs showed significantly higher rates of perifocal edema and ethmoid sinus infiltration compared to patients with all other midline meningiomas (Table 1), whereas patients with TSM showed significantly higher levels of optic nerve affection, bony infiltration of the planum sphenoidale, and anterior clinoid process (Table 2).

All median approaches were performed in patients with OGMs (frontal interhemispheric, n = 16; subfrontal, *n* = 2; bifrontal, *n* = 5), along with one patient with PSM (bifrontal, *n* = 1), whereas all other PSMs (*n* = 19) and all TSMs (*n* = 17) were resected via lateral approaches (pterional, *n* = 33; frontolateral, n = 28; supraorbital, *n* = 3). The tumor volume in patients who received resection via median approaches (MAs) was significantly larger compared to that of tumors approached via lateral approaches (LAs) (MA: 38.0 ± 38.3 cm^3^ vs. LA 12.8 ± 16.5 cm^3^; *p* < 0.001). Nevertheless, the extent of resection in tumors resected via median approaches (Simpson I 75.0%, Simpson II: 25.0%) was significantly better according to the Simpson grading compared to tumors resected via lateral approaches (Simpson 1 34.4%; Simpson 2 60.9%; Simpson 3: 4.7) (*p* < 0.001).

### 2.3. Functional Outcome

The median follow-up was 15.5 months (range 0–112 months), and the median postoperative KPSS was 90% (range 70–90%). Oculomotor nerve palsy appeared in 3.4% (3/88) (PSM 10.0%, 2/20; TSM 5.9%, 1/17) of patients and hemiparesis appeared in 1.1% (1/88) (OGM 2.0%, 1/51) of patients. Postoperative transient psychomotoric decline occurred in 6.8% (6/88) (OGM 9.8%, 5/51; PSM 10.0%, 2/20) of patients and hypopituitarism requiring lasting substitution in 2.3% (2/88) (TSM 11.8%, 2/17) of patients. Rates of psychomotoric decline were significantly higher after a frontal interhemispheric approach (OR 14.4; 95% CI, 2.3292-89.0260, *p* = 0.004). Preoperative visual deficits improved in 34.1% (30/88) of patients (OGM in 29.4% (15/51) of patients, PSM in 25.0% (5/20) of patients, and TSM in 58.8% (10/17)) (Table 3).

#### Adverse Events

The total rate of complications requiring surgical intervention was 17.0% (OGM 17.6%, PSM 20%, and TSM 11.8%). Postoperative epidural hematoma (EDH) necessitating revision surgery appeared in 3.4% of patients (OGM 2.0%, PSM 5.0%, and TSM 5.9%). Subdural hematoma (SDH) occurred in 4.5% (4/88) of patients (OGM 3.9% and PSM 10.0%). All EDHs and SDHs appeared in patients who underwent a prior pterional approach. Furthermore, 2.3% of patients (OGM 2.0%, PSM 5.0%) developed a cerebrospinal fluid (CSF) shunt-dependent hydrocephalus. CSF leaks appeared in 4.6% of patients (OGM 5.9%, TSM 5.9%). One patient with OGM and TSM was operated upon via a pterional approach, one OGM patient was operated upon via a bifrontal approach, and another patient with OGM was operated upon via a frontal interhemispheric approach. Tumor volume was a significant risk factor for developing a CSF leak (*p* = 0.002) and hydrocephalus (*p* < 0.001). Postoperative abscesses appeared in 2.3% of patients (OGM 3.9%) (Figure 1).

### 2.4. Median vs. Lateral Approach Regarding Operative Characteristics

Logistic regression analysis showed various significant parameters if a lateral approach was chosen: preoperative tumor volume (OR 0.966; 95% CI, 0.942–0.990, *p* = 0.006), preoperative KPSS (OR 1.051; 95% CI, 1.008–1.096, *p* = 0.020), a PSM (OR 15.607; 95% CI, 1.940–125.572, *p* = 0.001), a TSM (OR 6.160; 95% CI, 1.275–29.765, *p* = 0.024), an OGM (OR 0.034; 95% CI, 0.000–0.6398, *p* = 0.001), preoperative anosmia/hyposmia (OR 0.206; 95% CI, 0.760–0.557, *p* = 0.002), perifocal edema (OR 0.369; 95% CI, 1.378–0.988, *p* = 0.047), an affection of the ethmoid bone (OR 0.169; 95% CI, 0.387–0.741, *p* = 0.018), an infiltration of the cribriform plate (OR 0.232; 95% CI, 0.081–0.662, *p* = 0.006), tumor proximity to the anterior circulation (ACOMA, ACA) (OR 0.084; 95% CI, 0.008–0.871, *p* = 0.038) and infiltration of orbit/optic nerve/chiasmatic structures (OR 41.118; 95% CI, 1.503–1124.872, *p* = 0.028).

Based on these and previously mentioned findings, we developed an algorithm for choosing median and lateral approaches (Figure 2).

### 2.5. Exemplary Cases

#### 2.5.1. Exemplary Case One

A 41-year-old female patient presented with a headache and right visual decline. A **A** preoperative axial and **B** sagittal T1-weighted and gadolinium-enhanced MRI showed a TSM with chiasma and optic nerve affection. A **C** postoperative sagittal and **D** axial MRI control indicated a complete Simpson grade-I resection. The resection was performed through a pterional approach. Postoperatively, the patient did not recover from the 0.8 c. visual decline but remained stable during follow-up (Figure 3).

#### 2.5.2. Exemplary Case Two

A 47-year-old male patient presented with a headache, hyposmia, and lingering visual deficits. A **A** preoperative sagittal and **B** axial T1-weighted gadolinium-enhanced MRI showed an OGM with significant mass effect and perifocal edema. The preoperative tumor volume was 75.1 cm^3^. A **C** postoperative sagittal and **D** axial MRI control indicated a complete Simpson grade-I resection with no residual tumor. The resection was performed through a frontal interhemispheric approach. Postoperatively, the patient recovered from psychomotoric and visual decline. A galea-periosteum flap was used to prevent a postoperative CSF leak (Figure 4).

#### 2.5.3. Exemplary Case Three

A 68-year-old male patient presented with headache and fast progressing visual deficits. A **A** preoperative axial and **B** sagittal T1-weighted gadolinium-enhanced MRI showed a calcified PSM with mass effect. **A C** postoperative axial and **D** sagittal MRI control indicated a complete Simpson grade-I resection with no residual tumor. The resection was performed through a pterional approach. Postoperatively, the patient recovered from the visual deficits (Figure 5).

## 3. Discussion

In the present series, clinical and anatomical findings and the clinical follow-up were evaluated. Good clinical outcomes and high rates of Simpson I and II resections were achieved using standard skull-base approaches (Figure 1, Figure 2 and Figure 3). The treatment strategy in anterior skull-base midline meningiomas should always respect an individual’s anatomy, clinical presentation, and baseline characteristics. To help choose an appropriate approach, we propose a decision tree based on data from the present cohort. We believe the surgeon’s main choice is whether to access the lesion via a median or lateral approach. To do so, slight individualizations of the common “work horse” approaches (such as the midline or pterional craniotomies) can achieve good clinical results and resection extent [1,11,12,13,14]. This philosophy flattens the learning curve, harbors fewer risks than complex approaches, and might save surgical time compared to that used in complex approach preparation for the actual tumor resection.

### 3.1. The Median Approaches

In the present cohort, median approaches were mainly chosen for OGMs and PSMs. Median approaches might facilitate the intraoperative orientation because the anatomy is visualized in a straight line. The classic bifrontal approach allows good devascularization in OGM or PSM cases [7,8,14,15,16,17,18,19], but it makes visualizing the ACOM complex and optic nerves laborious, because they might be hidden behind tumor masses, thus rendering this approach less suitable for TSM patients [13,17,20]. Therefore, the interhemispheric approach might optimize control of the anterior vascular structures and optic nerves, which is why we advocate this technique. Preserving superficial veins is of utmost importance to reduce intra- and postoperative swelling of the frontal lobe. Adopted from the classic bifrontal approach, the transbasal/subfrontal approach enables superior devascularization of OGMs and PSMs with less brain retraction [6,7,17,21,22,23]. Nevertheless, the risk for CSF leakage might increase [4,7,17,24,25,26,27,28]. We also experienced this in patients with OGMs (Table 3). As an extension, the frontobasal interhemispheric approach via the interhemispheric fissure’s broader anatomical corridor might reduce frontal lobe retractions [7,23,29,30,31,32]. This technique offers a better visualization if the bony skull base is infiltrated [7,23,29,30,31,32].

### 3.2. The Lateral Approaches

The majority of patients (72.7% [64/88]) operated upon in the present cohort underwent a lateral approach. This group included pterional, frontolateral, and smaller supraorbital approaches. Lateral craniotomies allow excellent and quick visualization of the anterior circulation on both sides as well as superior access to the optic nerve, chiasm, and pituitary complex [1,11,12,13,14,17]. The classic pterional approach [11,12,16] offers a wide visualization of the parenchyma and the skull base, especially for lesions extending more laterally. However, the supraorbital approach is a minimally invasive alternative for more median-located lesions and for the ACOM complex [33,34]. Extradural preparation of the anterior clinoid process is advocated in cases with a more lateral TSM. For intradural preparation, meticulous drilling and irrigation is necessary to avoid heat-related damage to optic structures [4,6,17]. Various modifications of these “work horses” in skull-base surgery have been published and have trended in recent years [35,36,37,38,39].

### 3.3. Extent of Resection

For the vast majority of patients in this manuscript, an extensive resection was achieved according to the Simpson grading (96.6% achieved grades I and II). Comparable to Bassiouni et al., we reached significantly more Simpson grade-I results in OGMs (51.0% vs. 42.9%), even with infiltration of the anterior skull base or extension into the ethmoids or nasal cavity [19]. The resection extent did not correlate with initial tumor size in Bassiouni et al.’s series, indicating that a proper surgical approach may enable a total resection, irrespective of tumor extension. When the meningioma invades the frontal skull base, a median approach was chosen to optimize the visualization angle. Furthermore, preparing a wide frontal extracranial opening enables a large periostal flap for skull-base reconstruction or the use of a spilt bone graft technique.

### 3.4. Clinical Outcome

Clinical outcomes after resections of anterior skull-base midline meningiomas showed overall good results. A common main goal of resection is removing mostly benign tumor masses with slight or no neurological symptoms, so avoiding postoperative neurological deficits is of utmost importance. Our numbers align with prior reports, and comparisons of these individual skull-base lesions can be difficult [2,4,6,7,8,14,16,17,26]. Interestingly, clinical outcomes did not substantially differ when comparing the “classic” approaches in this cohort to more complex skull-base approaches. These differences bring into question their routine use and use for selected cases [9,21,40,41,42]. In the present cohort, interhemispheric and subfrontal approaches are associated with higher rates of relevant cognitive disorders after surgery, as controversially discussed in earlier publications [4,43,44,45]. Optic nerve and chiasm affections are often more safely handled from a lateral approach [13,16,19]. In the present cohort, postoperative visual improvement occurred in 35% of cases. Especially tumorous involvement of the orbit/optic structures emphasizes the posterior view’s advantage via the lateral approach, and the optic canal decompression can be done intra- or extradurally. The approach can occur from the side with more severely damaged optic structures. Regarding PSMs and TSMs, no clear consensus exists regarding the optimal approach. The frontolateral approach reportedly has a high rate of visual improvement (77.8%) [6].

Regarding huge infiltrating OGMs, preserving olfaction remains hardly possible. Various solutions have not substantially improved the outcome [4,7,8,9,26,29,46,47,48]. We did not see any postoperative improvement in our series.

In contrast, the outlook for visual improvement is better. In the present cohort, 25% of patients with PSM and 58.8% of patients with TSM had improved visual parameters, which aligns with prior reports on optic decompression [4,6,16,17,35,36,37,48].

### 3.5. Adverse Events

Comparing adverse events other than neurological deficits (see clinical outcomes), we identified a clear trend toward higher postoperative complication rates—especially for SDH and EDH—in lateral approaches. On the other hand, CSF leakage appeared more often in patients who underwent median approaches for OGMs, but still at rates comparable to previous reports [4,7,17,24,25,26]. This seems explicable, as OGMs infiltrating the thin cribriform plate tend to result in more CSF leaks and are regularly attacked via median approaches [27,28]. Overall surgical site infection rates are low (2.3% in the present cohort) and may be provoked by the larger tumor volume and longer surgery durations in our cohort [8,9,24,27,28,49]. Therefore, shorter surgery via a simple approach might impact infection rates, but this cannot be proven in this study.

### 3.6. Choosing a Suitable Approach

Based on the present cohort, we developed an algorithm to choose between median and lateral approaches. The most important factor is an OGM (*p* = 0.001) favoring a median approach or a relevant tumor mass effect (*p* = 0.006) requiring bilateral exposure or a direct perpendicular viewpoint. A lateral approach might be suitable to spare the anterior vascular complex or in the absence of bony arrosion of the skull base, in the absence of significant parenchymal oedema or in the presence of lower KPSS (*p* = 0.020). In more lateral lesions, such as TSMs and PSMs, a lateral approach is advocated. The median approach might work for patients without affection of the optic nerve or with a higher baseline KPSS and substantial parenchymal oedema.

Our findings suggest that a lateral approach’s advantages are emphasized in cases of PSM and TSM tumorous involvement of the optic structures. Early visualization of the visual apparatus is possible, brain retraction is reduced, and the exposure and angle of view are commonly known.

We do not differentiate between several techniques of median or lateral approaches and advocate using the most feasible approach according to institutional experience, habit and, of course, a tailored choice of approach for each case. Our proposed algorithm may serve as an additional tool, and we recommend keeping techniques as simple as possible.

### 3.7. Study Limitations

As a retrospective case series, we cannot draw causalities regarding clinical outcomes. Nevertheless, we implemented a detailed clinical examination, including scores on functional performance, and a standardized follow-up protocol based on a certified neurooncological board’s clinical workflow.

In addition, one should note that the impact of this manuscript could be seen as limited, as surgical approaches have been selected on an individual patient basis considering tumor and patient-related factors. As we decided to mainly focus on classic transcranial approaches and its technical considerations in order to obtain best oncological outcome, we did not include endoscopic approaches in our analysis. This point has to be discussed and may also be compared with classic approaches in future studies as minimal-invasive techniques are on the rise. In addition, the role of radiosurgery is often considered in treatment decision making of skull base meningiomas, especially in cases of patients not suitable for surgery or in case of recurrence or malignancy. Surgeon experience, prior training, and learning curves have to be taken into consideration as well.

In addition to its retrospective nature, the analyzed patient collective suffers from certain heterogeneities. For example, different types of meningiomas may create inhomogeneity. The number of patients with TSM and PSM was limited to 17 and 20, respectively, out of 88 total patients, which could lead to variability in the results. We included them in the analysis because all aspects of midline meningiomas of the anterior skull base should be reflected and because basic surgical techniques are similar for TSM and for PSM.

## 4. Materials and Methods

### 4.1. Study Design

We performed an observational retrospective single-center study. Patients who underwent surgery for midline meningiomas via transcranial approaches between June 2007 and January 2020 were included. Entities other than meningiomas were excluded from the analysis. The local ethics committee of the Technical University of Munich’s School of Medicine approved our study (231/20 S-EB). We conducted it in accordance with the ethical standards of the 1964 Declaration of Helsinki and its later amendments [50].

### 4.2. Outcome Parameters

The clinical records of patients who underwent surgery for a midline meningioma were analyzed regarding their preoperative neurological symptoms, Karnofsky Performance Status Scale (KPSS), new postoperative neurological deficits, postoperative complications, and re-interventions. OGM, PSM, and TSM were summed as midline meningiomas. The approaches used were dichotomized into lateral (pterional/frontolateral/supraorbital) and median (interhemispheric, bifrontal, subfrontal) approaches. The extent of resection was defined by comparing pre- and postoperative 3.0 T cranial magnetic resonance imaging (MRI) using T1 ± contrast agent sequences by manual volumetric segmentation, using the Origin^®^ software (Brainlab, version 3.1, Brainlab AG, Munich, Germany). Preoperative signs of calcification (in computed tomography) and perifocal oedema (3.0T MRI T2 sequences) were also noted.

### 4.3. Statistics

Statistical analysis was performed using the software package STATA version 13.1 (2011, StataCorp, College Station, Lakeway Drive, TX, USA). Normal distributions were assumed according to the central limit theorem. An unpaired 2-tailed Student’s *t* test was used to compare the significance of the means between two groups and were corrected for alpha error using the Holm–Bonferroni method. For the categorical variables, unpaired Mann–Whitney U tests were used to compare two samples. Proportions and group differences were analyzed with chi-square statistics. Additionally, logistic regression models with adjustment for potential predictive factors were calculated. Odds ratios with 95% confidence intervals (95% CI) were estimated. *p* ≤ 0.05 was considered significant.

### 4.4. Ethics Approval

The local ethics committee of Technical University Munich, School of Medicine, approved our study (231/20 S-EB). We conducted it in accordance with the ethical standards of the 1964 Declaration of Helsinki and its later amendments [50].

## 5. Conclusions

Even in large tumors, high rates of total tumor resection with few complications were achieved with simple approaches, allowing easy access with sufficient exposure. Considering the operative morbidity and functional outcome, the median approach provides satisfactory results in OGM cases, and lateral approaches enabled sufficient exposure of the optic apparatus in PSM and TSM cases. Proposing a simple decision-making algorithm for determining the best approach, the authors found satisfactory outcomes are possible for anterior skull-base midline meningiomas.

## Figures and Tables

**Figure 1 cancers-12-03243-f001:**
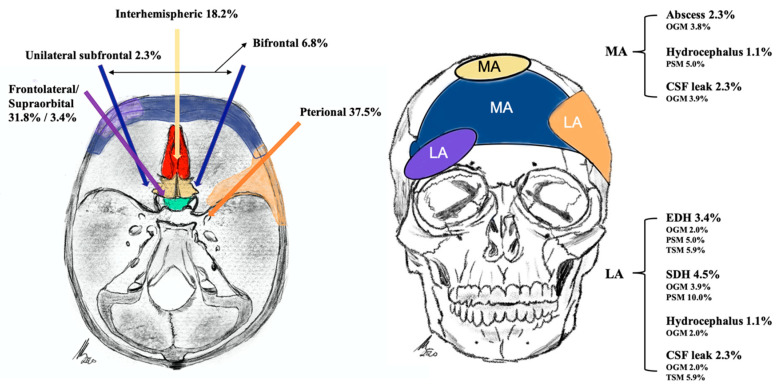
Illustration of performed approaches, angle of exposure, and approach-related complications, dichotomized into median (MA) and lateral (LA) approaches.

**Figure 2 cancers-12-03243-f002:**
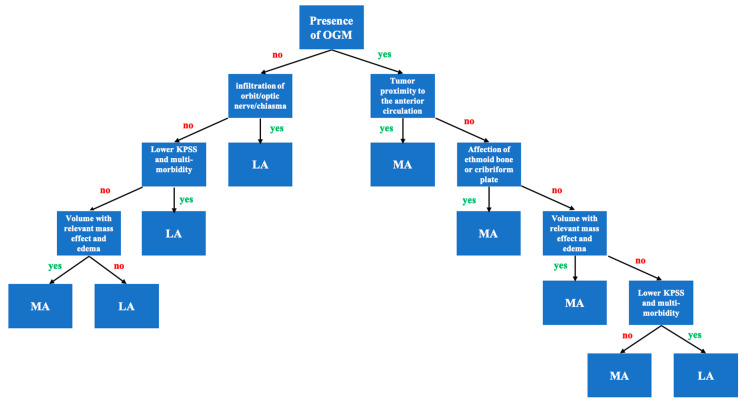
Proposed decision-making algorithm for targeting olfactory groove (OGM), planum sphenoidale (PSM) or tuberculum sellae meningiomas (TSM) based on our surgical experience, findings and statistical analysis. (MA = median approach; LA = lateral approach).

**Figure 3 cancers-12-03243-f003:**
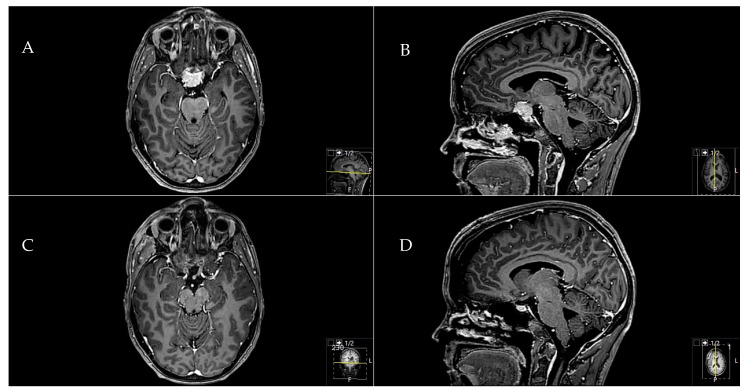
Exemplary case of a TSM with visual affection.

**Figure 4 cancers-12-03243-f004:**
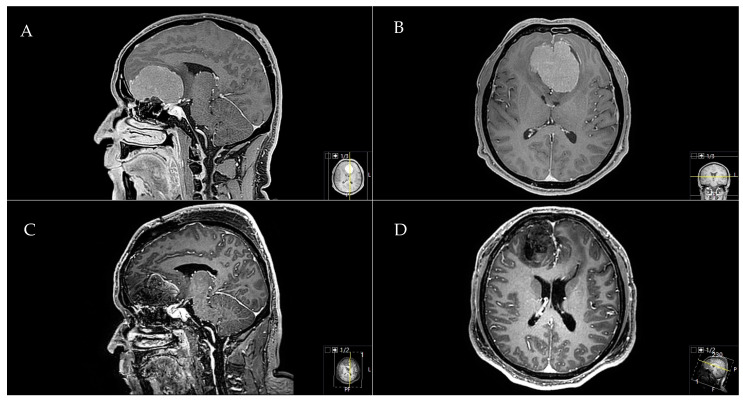
Exemplary case of a huge OGM.

**Figure 5 cancers-12-03243-f005:**
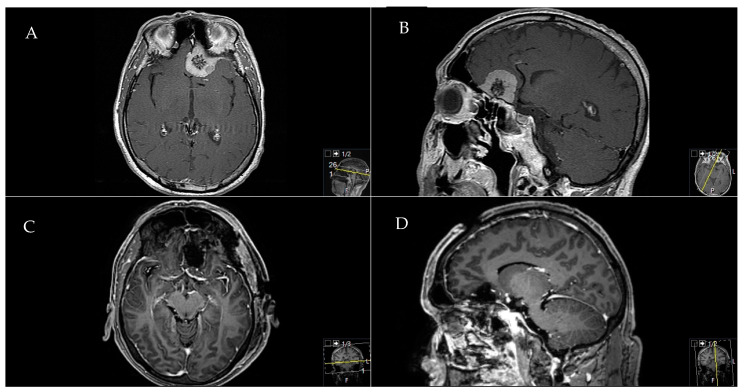
Exemplary case of a calcified PSM.

**Table 1 cancers-12-03243-t001:** Demographics and preoperative presentation.

		OGM (51)		PSM (20)		TSM (17)		Total (88)
Median (IQR), Mean (SD), N (%)			*p*		*p*		*p*	
Age (years)		60 (54–74)		62 (49–76)		53 (44–60)		60 (50–73)
Sex	♂	19 (37.3%)		5 (25.0%)		2 (11.8%)		26 (29.5%)
	♀	32 (62.7%)		15 (75.0%)		15 (88.2%)		62 (70.4%)
Preop. volume (cm^3^)		27 (±27)	0. 011	18 (±34)		3 (±2)		20 (±27)
Preop. KPSS (%)		80 (80–90)		90 (80–95)		90 (90–100)	0.027	90 (80–90)
Postop. KPSS (%)		90 (80–90)		90 (80–100)		100 (90–100)	0.016	90 (70–90)
Hyposmia/ Anosmia		51 (100.0%)	<0.01	2 (10.0%)		0		53 (60.2%)
Visual affection		13 (25.5%)		8 (40.0%)		10 (58.8%)	0.036	31 (35.2%)
Amaurosis		0		0		3 (17.6%)		3 (3.4%)
Psychomot. decline		17 (34.0%)	0.012	1 (5.0%)		0		18 (20.5%)
Vertigo		5 (9.8%)		0		0		5 (5.7%)
Seizure		9 (17.6%)		0		0		9 (10.2)
Hemiparesis		2 (3.9%)		0		0		2 (2.3%)

**Table 2 cancers-12-03243-t002:** Tumor characteristic and anterior skull base/neurovascular involvement.

	OGM (51)	*p*	PSM (20)	*p*	TSM (17)	*p*	Total (88)
**Calcification**	14 (27.5%)		5 (25.0%)		2 (11.8%)		21 (23.9%)
**Perifocal edema**	30 (58.8%)	**<0.01**	7 (35.0%)		0		37 (42.0%)
**Ethmoid bone/cribrose plate infiltration**	26 (51.0%)	**0.001**	1 (5.0%)		2 (11.8%)		29 (33.0%)
**Pituitary stalk deviation/affection**	6 (11.8%)		0		0		6 (6.8%)
**Frontal sinus/planum/sphenoid destruction**	9 (17.6%)		3 (15.0%)		5 (29.4%)	**0.037**	17 (19.3%)
**Orbit/optic nerve affection**	11 (21.6%)		9 (45.0%)		16 (94.1%)	**<0.01**	36 (40.9%)
**ACOMA/MCA/ACA involvement**	12 (23.5%)		6 (30.0%)		5 (29.4%)		23 (26.1%)
**Anterior clinoid infiltration**	0		2 (10.0%)		10 (58.8%)	**<0.01**	12 (13.6%)

**Table 3 cancers-12-03243-t003:** Postoperative functional outcome and complications.

	OGM (51)	PSM (20)	TSM (17)	Total (88)
Postoperative new visual deficits	4 (7.8%)	2 (10.0%)	0	6 (6.8%)
Visual improvement	15 (29.4%)	5 (25.0%)	10 (58.8%)	30 (34.1%)
EDH	1 (2.0%)	1 (5.0%)	1 (5.9%)	3 (3.4%)
SDH	2 (3.9%)	2 (10.0%)	0	4 (4.5%)
Hydrocephalus	1 (2.0%)	1 (5.0%)	0	2 (2.3%)
CSF leak	3 (5.9%)	0	1 (5.9%)	4 (4.5%)
Abscess	2 (3.9%)	0	0	2 (2.3%)

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
