# Peer review of "Midline Meningiomas of the Anterior Skull Base: Surgical Outcomes and a Decision-Making Algorithm for Classic Skull Base Approaches"

_cancers, 2020, doi:10.3390/cancers12113243_

Round 1

Reviewer 1 Report

Thanks for asking me to review this manuscript.

It addresses a useful question in a pragmatic manner, with a decent volume of cases analysed.

My comments:

1. Introduction: fair.

2. Methods: fair. This is placed, unconventionally, after the results, but the authors have followed the journal style as per instructions to authors.

3. Results: fair. The authors include adverse events, appropriately.

5. Figure 1: very good visual summary.

6. Illustrative cases very useful and realistic.

7. Discussion: very good and appropriate, including study limitations.

Author Response

Thank you very much for your proof-reading and your great review, we appreciate your interest in this manuscript. 

Comment 1: Introduction: fair.

Response 1: Thank you very much.

Comment 2: Methods: fair. This is placed, unconventionally, after the results, but the authors have followed the journal style as per instructions to authors.

Response 2: That is true, but we just followed the journal style instructions, but would, of course, change it if necessary.

Comment 3: Results: fair. The authors include adverse events, appropriately.

Response 3: And again, thank you very much for this review.

Comment 5: Figure 1: very good visual summary.

Response 5: Thank you very much for this opinion.

Comment 6: Illustrative cases very useful and realistic.

Response 6: Thank you, we just thought it would be nice to show some exemplary cases for better visualization.

Comment 7: Discussion: very good and appropriate, including study limitations.

Response 7: Reading this report makes us very happy and thank you for your kind comments.

Reviewer 2 Report

Summary: Researchers propose a simple decision-making algorithm for choosing suitable transcranial approaches for anterior skull-base midline meningiomas.

I accept this manuscript for publication

Author Response

Comment 1: Summary: Researchers propose a simple decision-making algorithm for choosing suitable transcranial approaches for anterior skull-base midline meningiomas.

I accept this manuscript for publication.

Response 1: Thank you very much for this report and review, we appreciate your opinion and would love to this manuscript published by Cancers.

Reviewer 3 Report

In this study authors present their experience with surgical resection of middle skull base meningiomas of various locations via different surgical approaches.

The novelty of this study is limited given that surgical approaches are selected on an individual patient basis considering tumor and patient related factors, overall treatment approach etc. The proposed algorithm takes into account tumor location and patient related factor. In addition, endoscopic approaches and also the role of radiosurgery is also often considered in treatment decision making of skull base meningiomas. Surgeon experience, prior training and learning curve also have to be taken into consideration.

Author Response

Comment 1: In this study authors present their experience with surgical resection of middle skull base meningiomas of various locations via different surgical approaches.

The novelty of this study is limited given that surgical approaches are selected on an individual patient basis considering tumor and patient related factors, overall treatment approach etc. The proposed algorithm takes into account tumor location and patient related factor. In addition, endoscopic approaches and also the role of radiosurgery is also often considered in treatment decision making of skull base meningiomas. Surgeon experience, prior training and learning curve also have to be taken into consideration.

Response 1: Thank you very much for this comment. We totally agree with you regarding above-mentioned limitations. As we decided to mainly focus on classic transcranial approaches and its technical considerations in order to obtain best oncological outcome, we did not include endoscopic approaches, also to maintain some homogeneity regarding the surgical techniques. Nevertheless, this point has to be discussed and may also be compared with classic approaches in future studies as minimal-invasive techniques are on the rise. Also, the role of radiosurgery, especially in cases of patients not suitable for surgery or in case of recurrence or malignancy plays an important role in tailored oncological therapy, as you correctly mentioned.

Your comment led us to expand our study limitation, as we think this may improve the quality and, from an oncological-surgical view, the impact of this manuscript for the interested readership.

We do hope the tracked changes in the revised manuscript are satisfactory enough regarding your correct concerns and we do hope for further consideration for publication. 

Thank you very much.

Reviewer 4 Report

This is a single-institution series of midline meningiomas operated via different surgical approaches. Authors present their experience and decision algorithm. The novelty of this report is not sufficiently clear considering that all patients were treated according to the best clinical practice guidelines. The manuscript could possibly be better suited for neurosurgery and/or skull base journal

Author Response

Comment 1: This is a single-institution series of midline meningiomas operated via different surgical approaches. Authors present their experience and decision algorithm. The novelty of this report is not sufficiently clear considering that all patients were treated according to the best clinical practice guidelines. The manuscript could possibly be better suited for neurosurgery and/or skull base journal.

Response 1: Thank you very much for this honest comment. Principally, you are right. Our decision algorithm is actually based on best clinical practice guidelines. Nevertheless, we thought our proposed algorithm may serve as an additional tool for choosing the most versatile and most feasible approach, but, as you correctly recognized, also according to institutional experience and habits. Of course, a tailored choice of approach for each case is of utmost importance as well. As to your knowledge, such simple illustrated algorithm is not published yet in case of classic approaches for such lesions, we saw the need to share our findings in order to possibly facilitate decision making. Indeed, our manuscript is emphasized on surgical considerations and techniques, but as meningiomas sill are oncologic lesions, we decided to submit it to your journal and would appreciate further consideration for publication.

Round 2

Reviewer 3 Report

I am fine with this new revised version.

Author Response

Thank you very much for this review and your helpful comments. The paper improved substantially through revision process.

Reviewer 4 Report

Nothing further to add. 

Author Response

Thank you for your efforts, the paper improved substantially through revision process.
Language has been edited by native speakers.